# Three-Dimensional Reconstruction Method of Rapeseed Plants in the Whole Growth Period Using RGB-D Camera

**DOI:** 10.3390/s21144628

**Published:** 2021-07-06

**Authors:** Xiaowen Teng, Guangsheng Zhou, Yuxuan Wu, Chenglong Huang, Wanjing Dong, Shengyong Xu

**Affiliations:** 1College of Engineering, Huazhong Agricultural University, Wuhan 430070, China; tengmore@webmail.hzau.edu.cn (X.T.); wuyuxuan@webmail.hzau.edu.cn (Y.W.); hcl@mail.hzau.edu.cn (C.H.); dwj@mail.hzau.edu.cn (W.D.); 2Key Laboratory of Agricultural Equipment for the Middle and Lower Reaches of the Yangtze River, Ministry of Agriculture, Wuhan 430070, China; 3College of Plant Science & Technology, Huazhong Agricultural University, Wuhan 430070, China; zhougs@mail.hzau.edu.cn

**Keywords:** three-dimensional reconstruction, ICP, Azure Kinect, RGB-D image processing, point cloud filtering, rapeseed

## Abstract

The three-dimensional reconstruction method using RGB-D camera has a good balance in hardware cost and point cloud quality. However, due to the limitation of inherent structure and imaging principle, the acquired point cloud has problems such as a lot of noise and difficult registration. This paper proposes a 3D reconstruction method using Azure Kinect to solve these inherent problems. Shoot color images, depth images and near-infrared images of the target from six perspectives by Azure Kinect sensor with black background. Multiply the binarization result of the 8-bit infrared image with the RGB-D image alignment result provided by Microsoft corporation, which can remove ghosting and most of the background noise. A neighborhood extreme filtering method is proposed to filter out the abrupt points in the depth image, by which the floating noise point and most of the outlier noise will be removed before generating the point cloud, and then using the pass-through filter eliminate rest of the outlier noise. An improved method based on the classic iterative closest point (ICP) algorithm is presented to merge multiple-views point clouds. By continuously reducing both the size of the down-sampling grid and the distance threshold between the corresponding points, the point clouds of each view are continuously registered three times, until get the integral color point cloud. Many experiments on rapeseed plants show that the success rate of cloud registration is 92.5% and the point cloud accuracy obtained by this method is 0.789 mm, the time consuming of a integral scanning is 302 s, and with a good color restoration. Compared with a laser scanner, the proposed method has considerable reconstruction accuracy and a significantly ahead of the reconstruction speed, but the hardware cost is much lower when building a automatic scanning system. This research shows a low-cost, high-precision 3D reconstruction technology, which has the potential to be widely used for non-destructive measurement of rapeseed and other crops phenotype.

## 1. Introduction

Computer vision can be used for target recognition and positioning [1], growth status diagnosis [2], agricultural products classification [3], yield prediction [4], agricultural machinery automatic navigation [5], phenotypic [6] and many other agriculture field. Computer vision technology can analyze complete phenotypic parameters such as plant structure, shape, color and texture at one time, being capable of to quantitatively study the growth laws of crops [7]. Three-dimensional (3D) reconstruction is a major research topics of computer vision. It performs digital modeling of crops in the computer, and keeps of 3D geometry and color of plant shape and organs in the computer to achieve rapid, low-cost, and fast agronomic traits of crops. Accurate non-destructive measurement [8]. This 3D phenotypic technique is meaningful for the research of crop breeding, growth and development, and biotic or abiotic stress [9]. At present, 3D reconstruction used in agriculture mainly include laser scanners, stereo vision, motion recovery structures, and RGB-D cameras. 

(1) Three-dimensional laser scanner. This is a high-precision point cloud acquisition instrument, but the scanning process is complex. It requires calibration objects or repeated scanning to accomplish the point cloud registration and stitching. The scanning process is usually time-consuming and difficult. In addition, the point cloud contains noise such as target objects and background, so point cloud segmentation becomes a challenge [10]. For example, Katrine et al. installed a 3D near-infrared laser scanner on an automatic boom with six different speeds, divided the scanning field into multiple sub-fields, calculated the depth profile of the XZ plane from each sub-field image, and calculated the plant height, projected leaf area, 3D leaf area and leaf angle. The distribution of the sub-field canopy was evaluated by using the laser scanner to perform high-time and high-resolution research on plants in the natural environment, but the maximum scanning width of the scanner and the distance between the flowerpots will limit the number of plants that can be measured in each sub-field [11]. Ana et al. used terrestrial laser scanner and mobile scanning system to obtain the point cloud of the vine trunk and calculated the volume of the vine trunk through a grape-shaped artificial object calibration method, and proposed a skeleton based on internal measurement. The algorithm for modeling a cylinder with a certain height and diameter effectively solves the problem of volume estimation. The relative error of different algorithms combined with different sensors is 2.46~6.40%. However, the two systems still have limitations such as long scanning time, cumbersome processing, and topography in the field [12]. Su et al. used six sets of laser scanners to obtain the corn point cloud in the whole greenhouse, by placing 10 high-reflectivity target balls, and ensuring that four target balls can be seen in each scanning position, using FARO SCENE 5.4.4 software for correction. The point clouds of different scanning positions were registered, the LIDAR360 software was used to reduce the point cloud noise points, the ground points and non-ground points were classified, and the ground point cloud was removed by normalization. This method supplies the possibility to obtain the overall corn point cloud. But the scanning time of each position scanner takes about 20 min, which are time-consuming and difficult [13]. 

(2) The structure from motion (SFM). This is a technology that can automatically recover the camera parameters and the three-dimensional structure of the scene from multiple image sequences. It has low-cost, high point cloud accuracy, and high color reproduction, but it is cumbersome and time-consuming to shoot sequence images. The three-dimensional reconstruction calculation takes a long time [14]. Liang et al. designed a crawler-type shooting trolley that can walk automatically according to a prescribed route. By controlling the trolley, it automatically collects corn plant images from different perspectives, uses Visual SFM for three-dimensional reconstruction, and uses statistical filtering for point cloud denoising and subsequent follow-up. The point cloud segmentation and phenotypic measurement simplifies the image acquisition work, but the position of the corn plants needs to be placed according to the distance, and the image shooting time is long and cannot solve the situation of dense plants [15]. Wei et al. developed an image acquisition system that places potted wheat on a rotatable platform driven by a servo motor and controls the rotating platform to take a set of digital photos of plants from 20 angles. This simplifies the image acquisition work and uses 2G-R-B vegetation. Index segmented wheat plants, and the average percentage errors of automatic plant height measurement and manual measurement were 2.71% and 10.06%, respectively. The processing time of each plant was greatly shortened, which provided the possibility to establish a high-throughput phenotyping platform [16]. 

(3) Stereo vision method. This uses two or more cameras to generate parallax from different perspectives, and then obtains the distance information of the object through deep calculation. The structure of the binocular camera is simple, and the calculation speed is fast, but it is greatly affected by the environment and is not suitable for scenes lacking texture information [17]. Xiong et al. built a semi-automatic image analysis system using binocular stereo cameras to extract leaf area and plant height. Report to manual measurement, the average error of the absolute percentage of leaf area and plant height measured automatically was 3.68% and 6.18%, respectively. However, the sample used is the seedling stage rapeseed with a minimal structure, and the complex morphology rapeseed plants in other growth stages have not been verified [18]. Rose et al. utilized a multi-view stereo method to reconstruct tomato plants, using two complementary perspectives to reconstruct the plant as a whole. Compared with the scanner, this method is more adaptable and saves space, but it still exists at the boundary of leaves and branches. Triangulation error, low point cloud color reproduction, and some point clouds abnormal values still need to be removed manually, which is not suitable for high-throughput collection work [19]. Chen et al. established two binocular vision systems and assembled them into a four-camera vision system. The positional relationship between the cameras was obtained by calibration. In the banana orchard with complex background, the banana images obtained by different cameras were semantically segmented to remove the background. It was proposed that the adaptive stereo matching algorithm can complete the stitching of multi-view point clouds with stable performance, which provides a reference for the 3D perception of bananas in complex environments. However, this article only performed partial three-dimensional reconstruction due to the camera perspective and failed to obtain an integral 3D information [20]. 

(4) RGB-D camera method. The RGB-D camera adds depth measurement to the function of the RGB ordinary camera. The mainstream solutions include structured light and time of flight. It is simple to operate, low-cost, and efficient. It has always had great potential in the field of target positioning and recognition and 3D reconstruction, but the quality of the point cloud is not high. There is a lot of noise and the problem of point cloud registration from multiple perspectives has always been the key issue [21]. Ma et al. used Kinect V2 to obtain the rape point cloud, used the minimum bounding frame to correct the angle between the rape point cloud and the camera, and combined a straight-pass filter to remove background noise, other outliers and floating noise point. The point cloud is removed based on the normal of the viewpoint and the surface, which improves the quality of the point cloud and the accuracy of the subsequent point cloud registration. However, the effect of this algorithm on the removal of floating noise point needs to be improved [22]. Xu et al. proposed a 3D scanning reconstruction system with multiple RGB-D cameras (Xtion sensors). The initial pose of the camera is obtained through a pre-calibrated image acquisition platform, and the coordinated scanning of multiple Xtion sensors can be achieved by the reconstruction system. The high-precision 3D model of complex and large scenes has good stability and robustness, but is limited by the problem of sensor accuracy, the reconstruction of small-sized objects has not been improved [23]. Hu et al. used Kinect V2 to collect the point cloud of lettuce plants at different growth periods and placed the plants in the center of the turntable. The background was black. The turntable was rotated every 20°. Plant data from 18 perspectives were collected, and 50 frames were continuously collected and averaged. In order to reduce random noise, the flowerpots and plants were divided according to the color information, the multi-view interference elimination algorithm was used to process the layered points, and then the moving least squares algorithm was used to reduce the remaining local noise around the leaf surface, using traditional the iterative closest point (ICP) registration algorithm registered the point clouds from different perspectives. The results show that the height and projected area of the lettuce measured by Kinect has a good linear relationship with the reference measurement value. The traditional ICP registration algorithm used by this algorithm takes a long time and the registration success rate is unknown [24]. 

In summary, the point cloud obtained by the laser scanner method has the highest accuracy, but it is expensive, cumbersome to operate, and complicated in subsequent processing. The SFM method has higher accuracy and the lowest cost, but high-precision 3D reconstruction requires a large number of images, which leads to the most serious consumption of computing resources. Stereo vision has the fastest reconstruction speed and the simplest structure, but its high-precision point cloud requires a high-performance camera and a stable imaging environment. The RGB-D camera represented by Kinect is more balanced in three aspects: point cloud accuracy, cost and processing speed. It has been more and more widely used in fields such as 3D reconstruction [22], target recognition [1], nondestructive measurement [25], plant phenotype [24] and so on. Regardless of which 3D reconstruction method is applied to plant phenotype detection, a registration algorithm is required to unify the point clouds from different coordinate systems into the same coordinate system to eliminate the occlusion and obtain a complete point cloud. In addition, point cloud segmentation and filtering and denoising are also necessary links. This is especially important for RGB-D sensors, because the point cloud data obtained is rough, outliers and floating noise point are serious [26]. 

Many scholars have explored the problems of point cloud registration, segmentation and filtering. Xu et al. used the Kinect V2 camera to take color images and depth images of rapeseed branches under four viewing angles and used the ultra-green segmentation algorithm to extract silique plants. After the calculation and removal of the largest connected domain, the rapeseed siliques were obtained. The quasi-algorithm is divided into coarse and fine registration, which greatly improves the registration speed and accuracy, and obtains a high-quality rape branch point cloud, but the point cloud noise generated by the camera is not removed well [27]. Vlaminck et al. Aimed at the impact of uneven point cloud density on point cloud registration, they proposed a multi-resolution point cloud registration method based on an octree. By reducing the number of points used to estimate the transformation and speeding up the nearest neighbor search, the algorithm greatly accelerates the registration speed and has strong robustness. However, the experimental objects in the article are all objects with small curvature changes and no color information. For crops, such as objects with large curvature changes It has not been verified [28]. Sun et al. proposed a Kinect V2-based high-throughput greenhouse plant point cloud reconstruction algorithm. Two RGB-D images on the turntable were collected to calculate the center point and normal vector of the turntable rotation axis, and the point clouds from different perspectives were rotated according to the center. Coarse registration, the iterative nearest point algorithm, is used to accurately register the multi-view point cloud to realize the rapid reconstruction of the three-dimensional point cloud of greenhouse plants. The algorithm uses the rotation center of the turntable for rough registration to make two point clouds over a large extent. The above this needs to be calibrated in advance before the image acquisition, the camera needs to be re-calibrated after the camera position changes, causing a lot of problems, and the reconstruction effect after ICP fine registration is not very good [29]. Through comparison with the traditional ICP registration algorithm, it was found that the algorithm’s accuracy and stability of point cloud registration have been greatly improved. The algorithm also has not been verified for the reconstruction effect of small-sized objects. In short, the research on 3D reconstruction based on RGB-D cameras mainly focuses on point cloud registration. In terms of accurate accuracy or speed, there is a lack of a method that takes into account both speed and accuracy, and there is a lack of powerful methods for point cloud filtering, especially for floating point filtering. 

The current mainstream RGB-D cameras mainly include the Realsense series produced by Intel, the Kinect series produced by Microsoft, and other companies’ products such as Xtion. The comprehensive performance of Kinect series RGB-D cameras is better than Realsense series in indoor environment. Azure Kinect is the latest model of the Kinect series. Compared with V2, the comprehensive performance has been greatly improved. There is still a lack of relevant in-depth research [30]. This paper proposes a three-dimensional reconstruction method using Azure Kinect and uses rapeseed as the object to test and verify the feasibility and performance of the method. This method collects color images, depth images, and near-infrared images from six viewing angles, improves the color-depth images alignment method provided by Microsoft to remove ghosting, and proposes a neighborhood extreme filter method to filter out the floating in the depth image. Then, the local point cloud is merged into an integral point cloud through the improved ICP registration algorithm. The point cloud denoising and registration enhance the performance of the algorithm, and finally realize the complete growth period of a single rapeseed. Accurate 3D color modeling: this 3D reconstruction method has high accuracy, high speed, low cost and easy automation. The point cloud quality meets the multi-scale, non-destructive, high-throughput, and high-precision measurement requirements of rape phenotypic traits. The research will provide important basic data for the non-destructive measurement of the rape phenotype, the study of growth law, selection and breeding, and the design of agricultural machinery, etc., and it can be extended and applied to other crops.

## 2. Materials and Methods

### 2.1. Experimental Setup and Data Acquisition

In 2019 and 2020, rapeseed was planted at Huazhong Agricultural University for two consecutive years, with varieties Zhongshuang 6 and Dadi 55. Rapeseed plants were removed in flowerpots before testing. The image acquisition device was mainly composed of Azure Kinect sensors, electric turntable, black screen and computer. As shown in Figure 1a, we placed the potted plant in the center of the turntable and placed a black backdrop behind it as a background. The Azure Kinect was aimed at the plant and formed an angle of about 15 degrees with the horizontal, 0.25~0.5 m away from the plant. The image acquisition method is shown in Figure 1b. The turntable was manually controlled to rotate 60° each time, and the computer controlled the Azure Kinect to capture images. A total of six viewing angles were captured. The image acquisition was carried out indoors with natural light. The image was taken using 4096 × 3072 resolution in color images, 1024 × 1024 resolution in depth images and 1024 × 1024 resolution in near-infrared images. The vision system was developed and tested with a personal computer. Its configuration was Intel Core i5-9300H/8G DDR4/256GB SSD/GTX1650 4G GDDR5. The operating system was Windows10 Professional. The required software was Microsoft Visual Studio 2017, OpenCV 3.2 and PCL1.8.1.

### 2.2. How to Obtain a Color Point Cloud in a Single-View

Taking rapeseed at the first flowering stage as an example, Figure 2 shows the algorithm flow of this manuscript. The color, depth, and near-infrared images of the plant at six viewing angles were acquired. We performed operations such as color-depth image alignment and point cloud smoothing on the image of each view, and then converted it into a color point cloud of a single view. The six color point clouds were merged into a integral point cloud by using the improved ICP iterative registration method, the point cloud filtering was performed again, and finally a high-precision color point cloud was obtained.

#### 2.2.1. RGB-D Image Alignment in a Single-View

RGB-D alignment is used to establish the index relationship of pixels in the color image and the depth image. Alignment is an important link for the point cloud to truly restore the color information. The software development kit (SDK) for Azure Kinect provided by Microsoft has an alignment function that can be used for alignment. However, there are ghosts in the results obtained by directly using this function to align, which requires optimization. We propose a mask method based on infrared binary image, which can achieve a good alignment effect. At the same time, this method removes the background in the depth image, which speeds up the conversion of a two-dimensional image into a three-dimensional point cloud and reduces the workload of subsequent point cloud segmentation. The process is shown in Figure 3 and mainly includes:

Step 1: using the alignment function in the SDK maps the color image to the depth image as the original result.

Step 2: reduce the near-infrared image from a 16-bit image to 8-bit image.

Step 3: utilizing the high reflectivity of the organism in the near-infrared image, using the basic global threshold method completely segments the plant from the near-infrared image, which may contain a small amount of background.

Finally: the binary image obtained in the third step is multiplied by the result obtained in the first step to obtain the final result.

#### 2.2.2. Color Point Cloud Acquisition in a Single-View

##### Floating Point Denoising

Floating point is a kind of noise phenomenon that is ubiquitous in sensors based on the time-of-flight principle. The light source will be refracted when it hits the edge of the object, causing the receiver to not receive the signal normally, resulting in a floating point. The floating points will gradually accumulate during the multi-view registration process, and have a significant effect on the reconstruction accuracy and computation speed. Therefore, this noise must be removed to improve the 3D reconstruction accuracy. We observe that the floating point has two distinctive features: first, it mainly occurs at the edge of the scanned object and the intersection of the front and back objects. Second, the floating point seriously deviates from its original position in space, which is reflected in the depth image as a sudden change in depth value. Using these two features, we propose a neighborhood extreme filtering method to smoothly filter the depth image to eliminate floating points. Including the following steps:

Step 1: binarize the near-infrared image to get an image that only contains the plant area, retrieve all non-zero pixels, and get the pixel point set$\;\Psi \left(x_{i},y_{i}\right)$.

Step 2: the depth image and infrared image taken by Azure Kinect has the same resolution. Traverse the pixels $\left(x_{i},y_{i}\right)$ with coordinates in the depth image, calculate the depth information $Z_{d}\left(x_{i},y_{i}\right)$ of a certain pixel $P_{d}\left(x_{i},y_{i}\right)$ in the neighborhood with the chessboard distance ≤N, and calculate the extreme value *Z_m_* (including the maximum value *Z_max_* and the minimum value *Z_min_*).

Step 3: calculate the value $\left|Z_{d}\left(x_{i},y_{i}\right)-Zm\right|$ and compare it with the given threshold value $Z_{s}$. If any value $\left|Z_{d}\left(x_{i},y_{i}\right)-Zm\right|$ is greater than $Z_{s}$, then we judge this point as a floating point and delete it.

Final step: repeat the above operations until the entire depth image is traversed, and all floating points are eliminated.

Through comparative experiments, it was found that when N is 3 and $Z_{s}$ is 5, the effect of removing suspended points is the best. By contrast with the traditional point cloud denoising method, this method is performed before the point cloud conversion. Compared with traditional methods such as pass-through filtering, radius filtering and statistical filtering, the speed and accuracy of denoising are greatly improved. However, this method will remove some non-floating points, causing damage to the point cloud accuracy, but the subsequent multi-view point cloud registration will repair this loss.

##### Color Point Cloud Conversion

The Azure Kinect sensor has system noise and fluctuation when it is working, which causes the data of each depth image to be different (which causes differences between each depth image data). It continuously shoots three frames of depth images and takes the average value to effectively eliminate fluctuations and noise. For the depth image without floating points, the function provided by the Microsoft SDK is used to directly convert the depth image into a point cloud. Then we add the color index information of the improved color-depth alignment result to the point cloud to obtain a color 3D point cloud. The conversion is shown in Equation (1).
(1)$$\{\begin{array}{l} X_{d}=\left(x^{\prime }-u_{OIR}\right)\times depth\left(x^{\prime },y^{\prime }\right)\times \frac{1}{f_{UIR}}\\ Y_{d}=\left(y^{\prime }-v_{OIR}\right)\times depth\left(x^{\prime },y^{\prime }\right)\times \frac{1}{f_{VIR}}\\ Z_{d}=depth\left(x^{\prime },y^{\prime }\right)/1000 \end{array}$$
in which, ($x^{\prime }$,$y^{\prime }$) are the image coordinate of a pixel in depth image, $\mathrm{depth}\left(x^{\prime },y^{\prime }\right)$ are pixel value of the ($x^{\prime }$,$y^{\prime }$), ($X_{d}$, $Y_{d}$, $Z_{d}$) are coordinate of 3D space, ($f_{UIR}$,$f_{VIR}$) are the focus length of the near infrared (NIR) camera.

##### Point Cloud Filtering

The point cloud obtained by the Azure Kinect camera contains many noise points. There were three types of noise in the point cloud: the background noise (BN), which consisted of non target points away from the targets; the flying pixel noise (FPN) from the boundaries of two objects, and the outlier noise, which consisted of scattered points, mostly around the targets, caused by the sensors [16]. The background point cloud has been well filtered out in Section 2.2.1. Suspended spots can be removed well by the method of Floating Point Denoising. The viewpoint feature and normal feature of the point cloud were used to remove the noise based on the spatial topological relationship established by the k-dimensional (kd) tree. The spatial topological relationship was used for searching neighboring points. With the statistical filter the point removal effect is general, mainly because the setting of the threshold parameter of the statistical filter is not easy to determine, the robustness is poor, and it is difficult to find a parameter that has a good effect on the two noise point clouds.

### 2.3. Registration for Multi-View Point Clouds

By registering and fusing point clouds from multiple angles into a whole, the integral 3D shape of the rape plant can be obtained, and the local information loss caused by the point cloud denoising operation can be repaired at the same time. The morphology and structure of rapeseed plants are complex, and the classic ICP algorithm is directly used for point cloud registration, which requires a large amount of calculation, is time-consuming, has a high error matching rate, and may fall into a local minimum. This paper proposes an improved method based on the classic ICP algorithm. By continuously reducing the distance threshold between corresponding points and the size of the grid during down sampling, the point clouds of the two perspectives are gradually merged into a high-precision point cloud. Repeatedly using this method for multiple viewing angles, a complete point cloud of a rape plant can be obtained.

#### Registration Optimization Based on ICP Method

(1) Point cloud registration from two perspectives

In the ICP algorithm, it is assumed that there are *N_P_* and *N_q_* points in the point cloud data *P* and *X*, which can be represented by point sets$\;\left\{P_{i}\right\}\;$and$\;\left\{X_{i}\right\}$. The ICP algorithm finds the two closest points in the point set and minimizes the sum of the squares of the Euclidean distance, as shown in Equation (2).
(2)$$f\left(q\right)=\frac{1}{N_{P}}\overset{N_{P}}{\underset{i=1}{\sum }}\|x_{i}-R_{\left(q_{R}\right)}p_{i}-q_{T}\|$$

In order to minimize f(q), the center of gravity of the point cloud *P* and the point cloud *X* are calculated separately, as in Equation (3).
(3)$$\{\begin{array}{l} \mu_{p}=\frac{1}{N_{p}}\sum_{i=1}^{N_{p}}p_{i}\\ \mu_{x}=\frac{1}{N_{x}}\sum_{i=1}^{N_{x}}x_{i} \end{array}$$

Use the center of gravity to obtain the cross-covariance matrix of the two point clouds, as in Equation (4).
(4)$$\Sigma_{P_{X}}=\frac{1}{N_{p}}\overset{N_{p}}{\underset{i=1}{\sum }}\left[\left(p_{i}-\mu_{p}\right){\left(x_{i}-\mu_{x}\right)}^{T}\right]=\frac{1}{N_{p}}\overset{N_{p}}{\underset{i=1}{\sum }}\left[p_{i}x_{i}{}^{T}\right]-\mu_{p}\mu_{x}{}^{T}$$

Construct $A_{ij}={\left(\sum_{p_{x}}-\sum_{p_{x}}^{T}\right)}_{ij}$ from the anti-symmetric matrix of $\Sigma_{P_{X}}$, construct $\Delta =\left[A_{23}A_{31}A_{12}{}^{T}\right]$, and then obtain the Equation (5) (4×4) symmetric matrix, $I_{3}$ (3 × 3) constant matrix:(5)$$Q\left(\Sigma_{p_{x}}\right)=\left[\begin{array}{cc} tr\left(\Sigma_{p_{x}}\right) & \Delta^{T}\\ \Delta & \Sigma_{p_{x}}+\Sigma_{p_{x}}^{T}-tr\left(\Sigma_{p_{x}}\right)I_{3} \end{array}\right]$$

Use the unit quaternion tree to represent the optimal rotation vector$\;q_{R}=\left[\begin{array}{llll} q_{0} & q_{1} & q_{2} & q_{3} \end{array}\right]$, which corresponds to the maximum eigenvalue of the matrix$\;Q\left(\Sigma_{p_{x}}\right)$. The rotation matrix can be calculated by Equation (6) and $q_{R}$:(6)$$R=\left[\begin{array}{ccc} q_{0}^{2}+q_{1}^{2}-q_{2}^{2}-q_{3}^{2} & 2\left(q_{1}q_{2}-q_{0}q_{3}\right) & 2\left(q_{1}q_{3}+q_{0}q_{2}\right)\\ 2\left(q_{1}q_{2}-q_{0}q_{3}\right) & q_{0}^{2}-q_{1}^{2}+q_{2}^{2}-q_{3}^{2} & 2\left(q_{2}q_{3}-q_{0}q_{1}\right)\\ 2\left(q_{1}q_{3}-q_{0}q_{2}\right) & 2\left(q_{2}q_{3}+q_{0}q_{1}\right) & q_{0}^{2}-q_{1}^{2}-q_{2}^{2}+q_{3}^{2} \end{array}\right]$$

From Equation (7), the point cloud transformation matrix is iterated repeatedly until the sum of squares of the Euclidean distance of the nearest point pair converges under a given threshold.
(7)$$q_{T}=\mu_{x}-R\left(q_{R}\right)\mu_{p}$$

Based on the classic ICP algorithm, this paper proposes an improved method, as shown in Figure 4a, the process is as follows:

Step 1: construct an effective initial point set participating in the ICP algorithm. Knowing the rotation angles of two adjacent viewing angles, the point cloud of one of the viewing angles can be multiplied by a rotation matrix to make the point cloud of this angle rotate around the Y axis to obtain a new point cloud, and then perform registration as in Equation (8).
(8)$$P_{R}=\left[\begin{array}{ccc} \cos\theta & 0 & \sin\theta \\ 0 & 1 & 0\\ -\sin\theta & 0 & \cos\theta \end{array}\right]\times P$$

This processing makes the point clouds of two adjacent angles similar in azimuth, which can effectively speed up the registration speed and success rate. In this manuscript, six uniform viewing angles are used. Therefore *θ* is 60 degrees. We calculate the error measure f(q) at this time.

Step 2: down-sample the point cloud to create a three-dimensional voxel grid, and use the center of gravity of the voxel grid to approximately replace all points for subsequent calculations. This can effectively improve the registration speed.

Step 3: calculate the surface normal vector and curvature of the point cloud, and search for points with similar curvatures to form a pair of points. On this basis, kd-tree is used to establish a high-dimensional index tree data structure, which accelerates the matching process of the corresponding points of the two point clouds.

Step 4: the distance threshold $D_{p}$ is used as an initial parameter to determine the search range of the registration process. Set $D_{p}$ to f(q)/10, start classic ICP registration.

Step 5: calculate a new error measure $f^{\prime }\left(q\right)$, If $f^{\prime }\left(q\right)$ is greater than the set registration error threshold, set $D_{p}$ to f(q)/20, and reduce the voxel grid size by 1/2 to release more points to participate in the registration. Perform classic ICP registration again.

Final step: keep $D_{p}$ unchanged, reduce the voxel grid 1/2 again, and perform the last classic ICP registration.

(2) Point cloud registration from all perspectives

In this manuscript, six perspective images of rapeseed plants were taken. The point cloud registration process for all perspectives is shown in Figure 4b. Name the point cloud of the first angle of view “Cloud 1”, and name the point clouds of the other 5 angles of view separated by about 60° as “Cloud 2”, “Cloud 3”, “Cloud 4”, “Cloud 5”, “Cloud 6”. First, perform the improved ICP registration method on the six-angle point clouds in sequence to obtain three new point clouds. For the point cloud 7 and the point cloud 8 rotated by 120 degrees, the point cloud 10 is obtained using an improved ICP registration method. For the new point cloud 10 and the new point cloud 9 rotated by 240 degrees, the improved ICP registration method is used to obtain the final completion point cloud.

## 3. Results

The experiments were carried out on raw data obtained from 40 pots of rapeseed, in which there were 10 pots in each growth period. For each pot of plant, six frames of data from different views, which cover 360°, were processed by the proposed method to show the performance and robustness of the proposed method. 

### 3.1. RGB-D Image Alignment Test

Figure 5 shows the processing results of a set of rape moss images. Figure 5a is the result of using the official SDK to align the flowering rapeseed and converting it into a point cloud and pass-through filtering to eliminate the background. Figure 5b is the alignment result and point cloud obtained by the proposed method. The alignment results and the point cloud in the comparison images Figure 5a and b have no ghosting. Figure 5c,d are the same type of test results of rape moss. We can find that for rape plants of different growth periods, the suggested methods can remove the ghost images in the alignment image, and also remove the noise of the point cloud.

### 3.2. The Influence of Shooting Background and Distance on 3D Reconstruction

In order to explore the impact of the shooting environment on the quality of the 3D reconstruction, the rapeseed silique branch was taken as the experimental object, and the rape branch images were taken under three conditions: indoor black screen, indoor white wall and outdoor environment. The shooting distance is 0.5 m. We used this method to process RGB-D images in three environments, and compare and analyze experimental results. The results are shown in Figure 6. It can be clearly seen from the figure that the missing point cloud is the least under the indoor black background. The number of silique point clouds was about 9838 under the black background, 6378 under the white background, and 7974 under the outdoor background. In general, the indoor black curtain was the best shooting environment. 

On this basis, the influence of shooting distance on 3D reconstruction was further explored. The best imaging distance within the imaging range (0.25–2.21 m) of Azure Kinect was tested with the black curtain indoors as the background. In the range of 0.25–1.5 m, RGB-D sequence images were taken every 0.25 m, the images were processed by the method in this paper, and the experimental results were compared and analyzed. The branch point cloud of rape was obtained at these six distances. As a result, by adapting our method as shown in Figure 7, we can find that when the shooting distance is greater than 0.5 m, the point cloud quality decreases significantly. The point cloud quality is the best when the shooting distance is 0.25 m, but a complete image cannot be obtained for objects with larger sizes at this distance. Therefore, the proper shooting distance should be set between 0.25–0.5 m according to the size of the object. 

### 3.3. Point Cloud Noise Removal Test

We tested and compared two filtering methods. The original point cloud as shown in Figure 8a. The traditional method of pass-through filtering and statistical filtering, as shown in Figure 8b. The recommended method of neighborhood extreme filtering and pass-through filtering, as shown in Figure 8c. The point cloud directly converted from the depth image has about 770,000 data points, most of which are non-target noise points. This part of the noise point cloud is far away from the target crop and can be easily removed from the through filter. It can be seen from figure that the pass-through filtering can remove the noise point cloud well in a large area such as the background. Comparing the filtering effect identified by the red dashed box, both methods can well remove the floating points with obvious characteristics. However, at the position marked by the yellow dashed frame, which is the floating point caused by the overlap of the leaves, the filtering effect of statistical filtering is almost invisible, and the method in this paper has achieved remarkable results.

As shown in Figure 9, We tested the rape plants in four growth stages (seedling stage, moss stage, flowering stage and silique stage), with 10 samples in each stage. We counted NAF (the mean number of filtered point clouds), NRR (noise reduction ratio), and time-consumption. Figure 9a compares the performance of the two filtering methods. Compared with the traditional method, the suggested method is about 1378 points less than the traditional method in terms of the number of point clouds. The average NRR of the proposed method is 0.305% higher than that of the traditional method. This is reflected in the filtering effect, and the proposed method is more thorough in filtering noise. There is a significant difference between the two in terms of time-consumption, and the traditional method is 11.15 times that of the suggested method. The method in this paper benefits from the noise filtering before the point cloud conversion. The two-dimensional image processing saves a lot more time than the three-dimensional point cloud processing, which reflects the time-consuming. In addition, the speed of pass-through filtering depends on the number of point clouds. The number of point clouds involved in pass-through filtering in this method is significantly less than that of traditional methods, which in turn promotes further reduction in time-consuming. In general, the method in this paper can eliminate point cloud noise very well, and has significant advantages in terms of time-consumption.

The optimal filtering method needs to set an appropriate neighborhood size and threshold. The size of the neighborhood is too small to judge the sudden change of the depth information, which leads to the deletion of too many valid points. However, too many points will participate in the judgment if the setting is too large, so that the deep mutation information is inaccurate and the mutation point cannot be found. Through the statistics of the gentle point cloud of a certain part of rape, it is found that the maximum depth difference in the local neighborhood is not obvious, and the difference is basically 5 points. This paper compares the size of different neighborhoods and the size of the threshold to eliminate the effect of floating point noise, and selects the final parameters. Figure 10 shows the filtering results of different parameters, the neighborhood sizes are 3 × 3, 5 × 5, and 7 × 7, and the thresholds range from 3, 5 to 7. Through comparison, it is found that when the neighborhood size is 3 × 3 and the threshold is 5. The effect of removing the noise point cloud is the best.

### 3.4. Point Cloud Registration Experiment

We tested and compared the performance of the classic ICP method and the method in this paper for the two-view point cloud registration, The classic ICP registration algorithm is only one registration and no parameters are changed in the process. which is shown in Figure 11 and Figure 12. For any rape plant, the method in this paper undergoes three iterations of registration, and finally merges into a point cloud precisely. Figure 11 clearly shows the gradual fusion of the two point clouds. Compared with Figure 11a using our method, Figure 11b shows the result using the traditional method, the registration effect of the classic ICP method is much worse and the point clouds of rape seedlings are not well integrated. For two point clouds with large differences in angles, the registration may even fail, as shown in Figure 11d. Figure 11c,d are the same type of test results of rape seedlings. Figure 12 compares the registration success rate and time-consumption of the two methods. Forty sets of test results show that the success rate of traditional ICP registration is 32.5%, and the success rate of this method is 92.5%. The average time consumption of the traditional ICP method is 213 s. Although the method in this paper includes three ICP registrations, the average time consumption is 65.6 s, which is only 30.8% of the traditional ICP method. The substantial increase in registration efficiency is mainly attributed to the optimization measures taken. The rotation processing of the original point cloud greatly shortens the effective point cloud distance of the two point clouds, which significantly reduces the probability of falling into a local optimal solution, improves the registration success rate, and also increases the registration speed. The OcTree search method improves the search speed. Multiple iterations of registration make the process of fusion of the two point clouds keep improving, and finally achieve a more perfect fusion. In general, compared with the traditional ICP registration algorithm, the registration algorithm in this paper has a significant improvement in registration performance, and can accurately reconstruct rape plants.

### 3.5. Integral 3D Reconstruction Experiment

The point cloud obtained by the laser scanner is extremely accurate, and has become a benchmark reference for comparison experiments in 3D reconstruction. The laser scanner used in this manuscript was a WibooxPSC-TAAJ011A15 produced by Xianlin 3D Technology Co., Ltd. The scanner and the color camera were fixed to the robot arm (model AUBO ROBOTS) together, and the data of 6 angles of view of the rape plant were taken for 3D reconstruction [31]. The color 3D point cloud obtained was compared with the method in this paper. Figure 13a shows the working site and reconstruction results of the color laser scanner. As can be seen from the details marked in the figure, in terms of spatial resolution and color reduction, the laser scanner surpasses the method in this paper, and the point cloud of the stem part exceeds the method in this paper, but the point cloud of the leaf and flower part where the curvature changes greatly exceeds that of the laser scanner. The overall reconstruction effect of the laser scanner is better, especially the continuous and smooth parts of the reconstruction quality are excellent. However, the effect is not good for the parts with sudden curvature changes such as leaves and stems, and the point cloud is prone to defects and holes. In the partial enlarged images, only a small part of the point cloud is recovered from rape blossoms, which are almost impossible to reconstruct. Figure 11b shows the reconstruction results of this method for the same rapeseed plant. In all parts of the rape plant, it can be rebuilt very well with very few holes. Figure 13c,d are the contrast effects of rapeseed in the moss stage. It can be clearly observed that the leaves reconstructed by the three-dimensional scanner have a lot of authenticity, and the suggested method obtains a complete leaf point cloud. In general, the laser scanner can obtain more precise details, and the suggested method can obtain a more integral point cloud. At the same time, the method proposed in this manuscript is worse than the 3D laser scanner in terms of visual effects. This is mainly due to the weaker performance of the RGB camera used by Azure Kinect, which leads to insufficient color reproduction.

Figure 14 compares the reconstruction accuracy and speed of the two methods. The reconstruction accuracy of the method in this paper was 0.789 mm, and the 0.191 mm of the laser scanner was sub-millimeter reconstruction accuracy. The average time for a integral 3D reconstruction of the method in this paper was 302 s, while that of the laser scanner was 901.4 s. The former is 1/3 of the latter. If an automated image acquisition device is used, the scanning method of Azure Kinect will be faster. In general, the reconstruction accuracy of this method is close to that of a laser scanner, but the speed is 3 times faster. This is mainly due to the fact that the data collection speed of the laser scanner and the point cloud fusion speed are not as good as the Azure Kinect method. In terms of the 3D reconstruction technology required for phenotypes, the laser scanner equipped with a robotic arm can realize an automated 3D reconstruction system, but the cost of the system is very high (often more than $30,000). The method in this paper only needs to configure an electric turntable and assembly line, the structure is simple, the image acquisition efficiency is higher, and the price is very low (less than $1000). 

### 3.6. Analysis and Discussion of 3D Reconstruction of Rapeseed Plants

The method in this paper was applied to rapeseed in the w hole growth period, and the relevant experimental data are shown in Figure 5, Figure 6, Figure 7, Figure 8, Figure 9, Figure 10, Figure 11, Figure 12, Figure 13 and Figure 14. We analyzed the success rate and point cloud quality separately.
(1)Seedling stage. The rapeseed plants in this period are very short, and the camera can take images at a relatively close distance, so that the accuracy of the rape point cloud obtained is higher. But also because the plant is too small, the point cloud curvature characteristics are not obvious, so that the matching point pairs are too few and the registration is easy to fail.(2)Seedling stage and moss stage. The morphological differences of rapeseed plants at this stage are not obvious, but they are significantly larger than the seedling stage, so the shooting distance has to be increased to ensure that a complete plant image is captured, which makes the difference between the point cloud quality and the seedling stage insignificant. However, due to the richer morphological structure of rape at this stage, the registration success rate is greatly improved.(3)Silique stage. The plant type of rapeseed in the silique stage is tall (often more than 2 m in height), and it is necessary to cut the rapeseed into branches before shooting. The branch size is similar to the moss stage. However, the stalks and stems in this branch are very small (often less than 1 mm), which is close to the minimum spatial resolution of the method in this paper. At this time, the point cloud is often missing, the number of matching point pairs is sharply reduced, the traditional ICP method is almost unusable, and the registration success rate of the method in this paper has also decreased. All in all, the method in this paper can achieve a higher 3D reconstruction success rate and better point cloud quality for the whole growing period of rapeseed plants.

## 4. Discussion

For the application of Azure Kinect in the field of 3D reconstruction, this article provides a process and method with universal significance. For RGBD image sequences with multiple viewing angles, a colorful 3D point cloud is obtained through image preprocessing, point cloud preprocessing and point cloud registration. Targeted solutions are proposed for several problems in RGB-D 3D reconstruction. 

Firstly, to solve the problem of poor color image-depth image alignment, a comprehensive processing method based on RGB-D-NIR images is proposed. Compared with the official Microsoft alignment result, the ghosting has been eliminated and the color image has become more accurate. In addition, the invalid background 3D point cloud is greatly reduced at the same time, which improves the speed and accuracy of subsequent point cloud processing. 

Secondly, a neighborhood extreme filtering method is proposed to filter out the brightness mutation points in the depth image, which effectively eliminates the floating points and outliers, and improves the quality of the point cloud. Compared with the traditional method of pass-through filtering and statistical filtering, the method in this paper has achieved significant results in eliminating floating points. The number of point clouds is 76.5% of the traditional method. The average NRR is 0.269% higher than the traditional method, and the SD value is equal. The time-consumption is about 1/16 of the traditional method. 

The third and most important outcome is that an improved registration method based on ICP is proposed. Through multiple iterations, the key parameters in the registration are reduced, so that the point cloud is gradually approached in space, and finally a high-precision point cloud registration is realized. Compared with the traditional one-time ICP registration method with a lower registration success rate, this method has a success rate of 100%, and the time-consumption is only 1/3 of the traditional ICP method. 

Finally, the three-dimensional scanning system constructed by this method was applied to rapeseed plants during the whole growth period, and a satisfactory three-dimensional reconstruction effect was achieved. It has also achieved good effects on crops such as fruit seedlings, corn, soybeans, and cotton. Compared with expensive 3D laser scanners, the reconstruction accuracy is sub-millimeter level, the point cloud details have their own advantages, and the time-consumption is 1/3 of the laser scanner. All in all, the method in this paper proposes a creative solution in point cloud registration and filtering denoising, and it is also a useful example of 3D reconstruction research based on RGBD camera.

There is some potential for improvement of the presented method. Firstly, the performance of the color camera of the Azure Kinect sensor is weak. The color reproduction of the 3D reconstruction can be improved by installing a constant light source darkroom, color calibration and correction, or installing a higher-performance camera. Secondly, the calibration methods can be developed for the lens distortion of both the RGB and D camera, and distance measurement deviation of D camera. This work can improve accuracy in 3D reconstruction. Finally, the current method employed six views in a constant distance, which restrict a limited target size. Further research may resolve 3D reconstruction for a large-scale plant with more views.

## Figures and Tables

**Figure 1 sensors-21-04628-f001:**
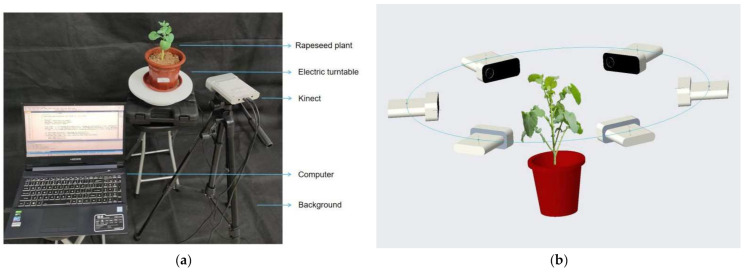
Rape plant image collection. (**a**) Image acquisition platform. (**b**) Collection diagrams.

**Figure 2 sensors-21-04628-f002:**
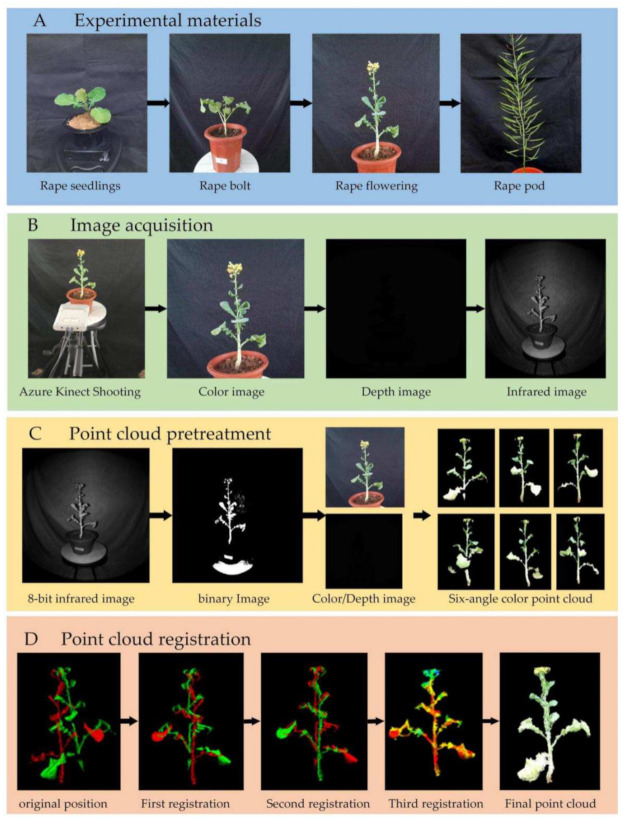
The program flow chart of this manuscript.

**Figure 3 sensors-21-04628-f003:**
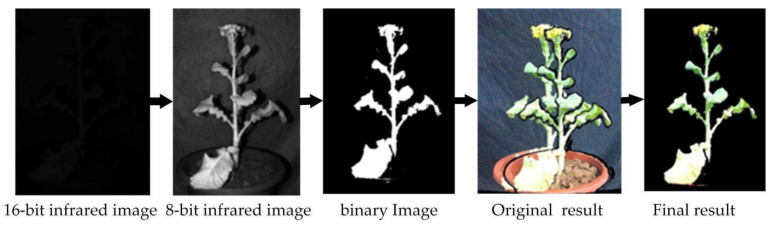
RGB-D images alignment.

**Figure 4 sensors-21-04628-f004:**
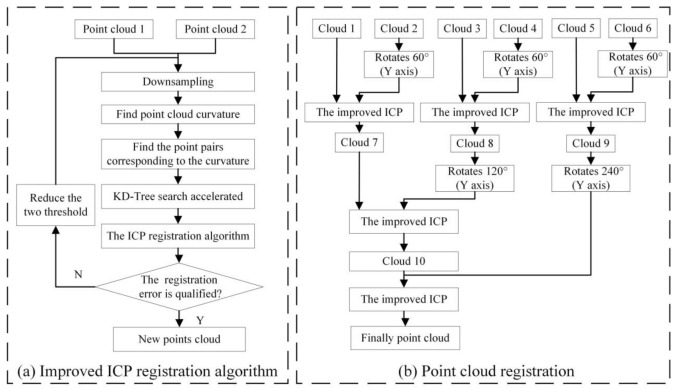
Flow chart of the point cloud registration. (**a**) Improved ICP registration algorithm. (**b**) Point cloud registration of six views.

**Figure 5 sensors-21-04628-f005:**
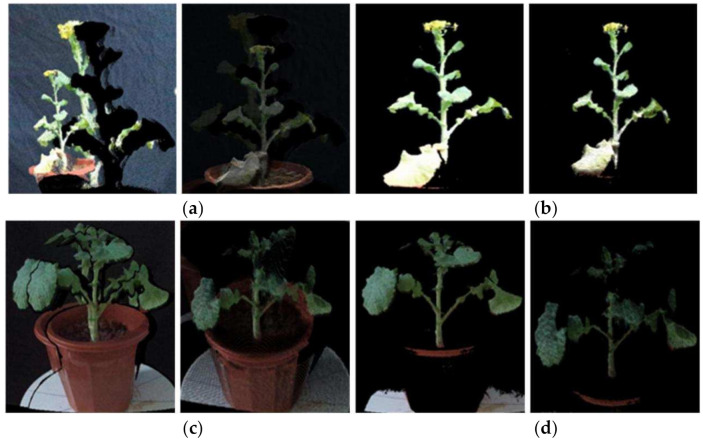
RGB-D image alignment test. (**a**) The alignment result and point cloud by using the official software development kit SDK. (**b**) The results by using the proposed method. (**c**,**d**) are the other examples of rapeseed moss.

**Figure 6 sensors-21-04628-f006:**
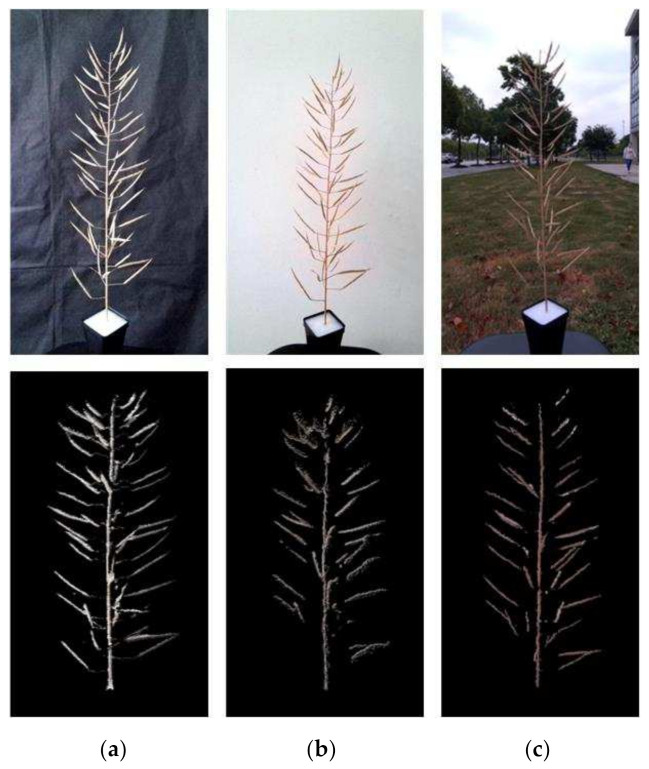
Comparison of different backgrounds (**a**) Indoor black background. (**b**) Indoor white background. (**c**) Outdoor background.

**Figure 7 sensors-21-04628-f007:**
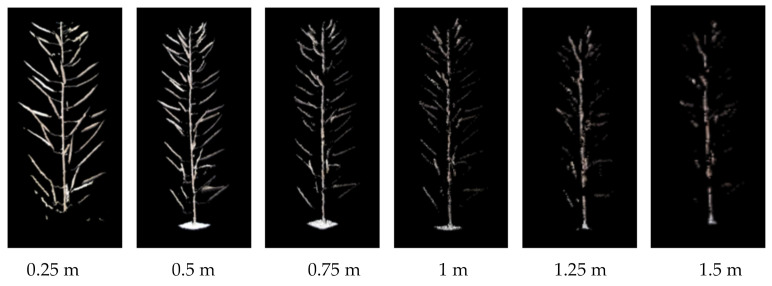
Comparison of silique point clouds at different shooting distances.

**Figure 8 sensors-21-04628-f008:**
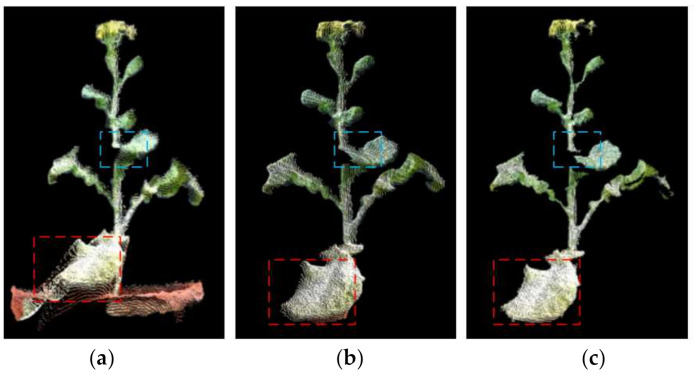
Comparison of two filtering effects. (**a**) The original point cloud. (**b**) Pass-through filtering and statistical filtering. (**c**) Neighborhood extreme filtering and pass-through filtering.

**Figure 9 sensors-21-04628-f009:**
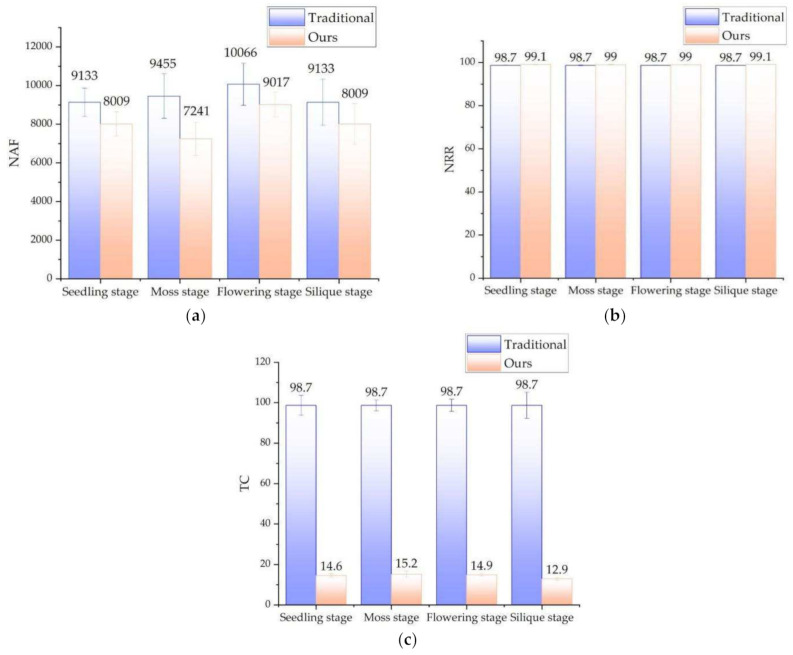
Comparison of the effect of the traditional method and ours. (**a**) comparison of the NAF. (**b**) comparison of the NRR. (**c**) comparison of the time consuming. Note: NAF: Means number of point clouds after filtering. NRR means noise reduction ratio, NRR = (1 − number of point clouds after denoising/number of original point clouds) ∗ 100. TC: Time consuming (Second).

**Figure 10 sensors-21-04628-f010:**
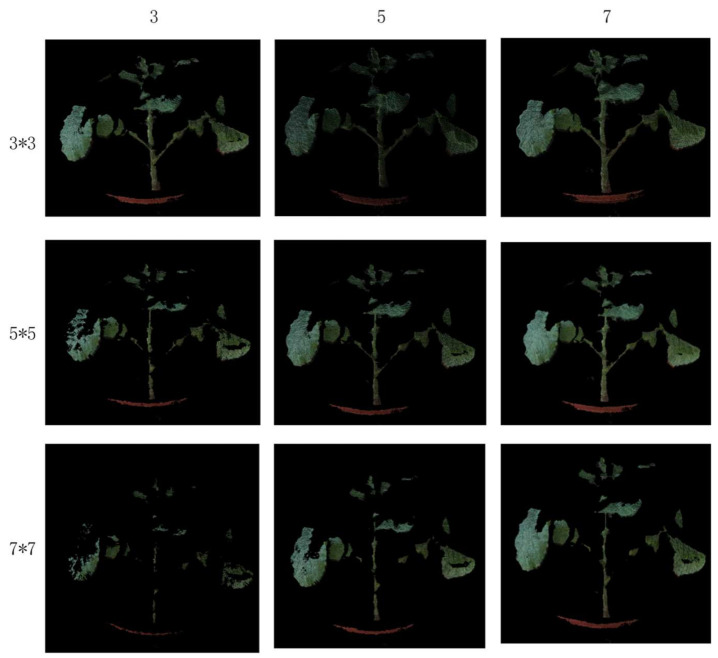
Comparison of the effect of different thresholds of the maximum filter.

**Figure 11 sensors-21-04628-f011:**
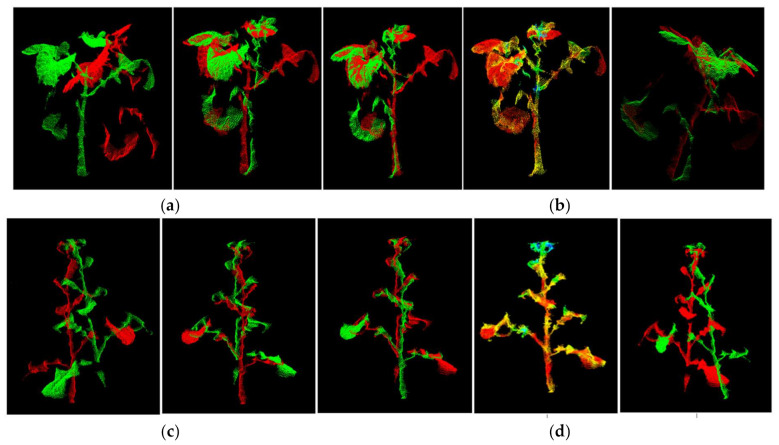
The registration effect of the traditional ICP method and the method in this paper. (**a**) The registration method of this paper. (**b**) Traditional ICP registration. (**c**,**d**) another set of registration data.

**Figure 12 sensors-21-04628-f012:**
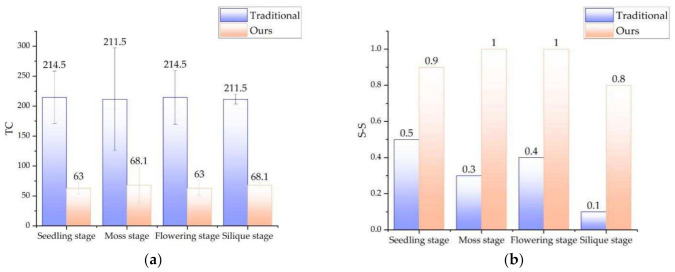
Comparison of the time and effect between the traditional registration algorithm and ours. (**a**) the comparison of the TC. (**b**) the comparison of the S-S. Note: TC: Time consumption (Second). S-S: succeed status (Yes or No).

**Figure 13 sensors-21-04628-f013:**
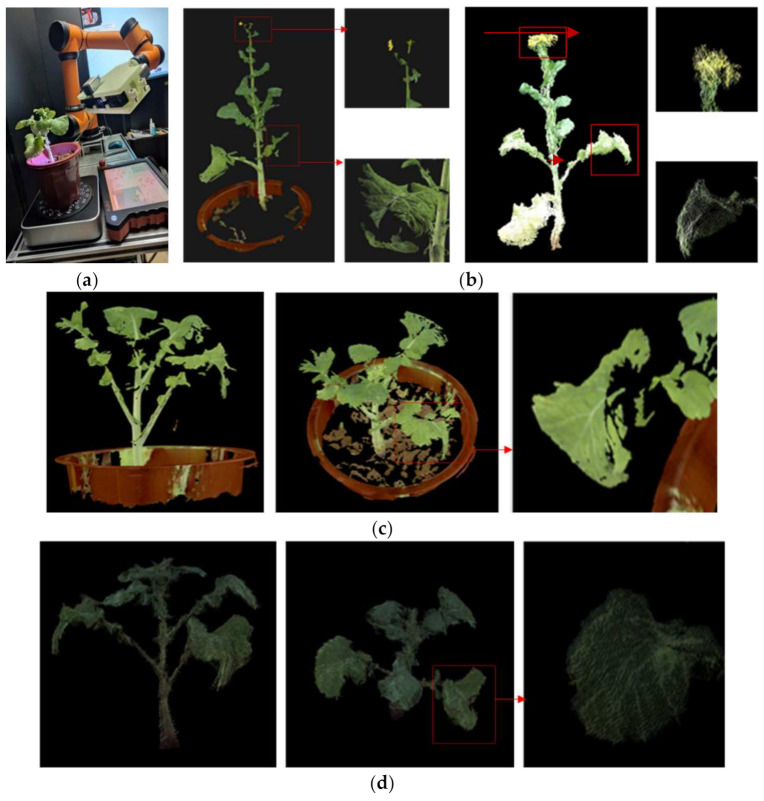
Comparison of the 3D reconstruction effect between the laser scanner and the algorithm in this paper. (**a**) 3D reconstruction of laser scanner (flowering rapeseed). (**b**) 3D reconstruction of our method using the same sample. (**c**) Another 3D Reconstruction of 3D laser scanner (Rapeseed in moss stage). (**d**) 3D reconstruction of our method using the same sample.

**Figure 14 sensors-21-04628-f014:**
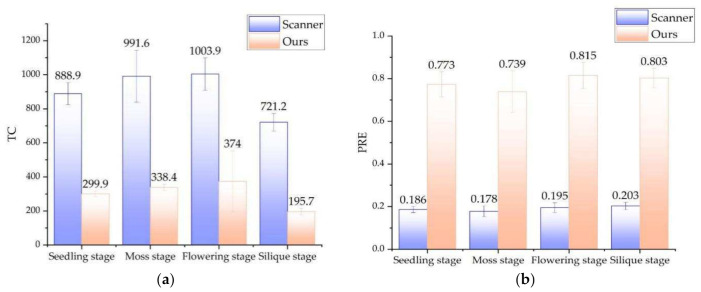
Comparison of resolution and time consuming between a laser scanner and our method. (**a**) Comparison of the TC. (**b**) Comparison of the PRE. Note: TC: Time consumption (Second), PRE: Precision (mm).

## Data Availability

The data used to support the findings of this study are available from the corresponding author upon request.

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
