# Peer review of "Three-Dimensional Reconstruction Method of Rapeseed Plants in the Whole Growth Period Using RGB-D Camera"

_sensors, 2021, doi:10.3390/s21144628_

Round 1
Reviewer 1 Report
Overall Decision
Major revision
In this manuscript, Azure Kinect sensor is used to shoot color images, depth images and near-infrared images of the target from six perspectives, and then point clouds are generated according to RGB-D camera. The reprojected images and most background noises are removed by point cloud filtering and denoising, and finally a complete point cloud image is generated by improved ICP method. This article has a certain degree of innovation, but the author has some doubts about writing style and content. It is hoped that the suggestions can improve the quality of manuscripts.
Chapter 1: Introduction
- The author has spent a lot of space describing the research status of point cloud generated by 3D laser scanner, SFM, binocular camera and RGB-D camera. However, I feel that it simply describes the research contents of the above methods by relevant researchers, and I hope the authors can supplement the defects of their respective research contents and their own solutions, instead of simply explaining the shortcomings and shortcomings of various sensors. For example, when other researchers use RGB-D cameras, the registration of RGB images and infrared images is a problem.
- The first paragraph introducing the research topic may present a much broaderview of the problems related to your topic and be revised and completed with citations to computer-vision based authority references, especially with RGB-D cameras applications (Recognition and localization methods for vision-based fruit picking robots: a review).
- The author has too many research contents on one sensor, so I hope the author can only select four representative research contents for each sensor to analyze.
- Vision technologyapplications in various engineering fields, should also be introduced for a full glance of the scope of related area. For object detection, please refer to fruit detection papers such as Fruit detection in natural environment using partial shape matching and probabilistic Hough transform. For 3D reconstruction, please refer to Three-dimensional perception of orchard banana central stock enhanced by adaptive multi-vision technology.
Chapter 2: Materials and methods
- The shortcomings of RGB-D cameras include that Infrared rays are greatly affected Under outdoor lighting conditions. In this paper, the author puts potted plants in the center of the turntable and puts a black background cloth behind it. This method may eliminate this error. The experiment under this specific background is not very convincing. I hope the author can supplement the experimental results and analysis under different background conditions.
Chapter 4: Results
- The reference numbers in fig. 6 are not aligned. Please revise it.
- In chapter 4.4, 10 groups of experimental data may not be enough to test the author to increase the number of experiments; The author is invited to consider the experiments of different distances between sensors and plants.
Chapter 4: Discussion and conclusions
- It is suggested to summarize the technical process, major contributions and data analysis conclusions of this paper in the conclusion part.
- The final conclusion needs to supplement some experimental data and comparative conclusions to prove the innovation of the proposed method.
Author Response
Thank you for your suggestions on our paper, which helped us a lot.
We have reorganized and edited the Introduction and conclusions. Thank you for your suggestions. I have learned a lot.
Regarding your suggestion to increase the shooting background and shooting distance, we have added this part of the experiment. Thank you for your suggestion.
10 sets of experimental data are indeed not enough to explain the problem. We have verified 10 rapeseed plants in each growth period, a total of 40 sets of data, I believe this will make the data more perfect.
The picture in the article has been carefully corrected, thank you for your careful reminder.
Thank you again for your valuable suggestions.

Reviewer 2 Report
The paper provides a couple of faster techniques for noise removal, registration, and stitching of 3D point clouds of Rapeseed plants (in a controlled environment) collected using an Azure Kinect sensor. Kinect sensors are a less expensive alternative to laser scanners, and I believe that this type of reconstruction work is always beneficial to the border plant scientific community.
- The authors proposed a ‘multiple iteration' ICP algorithm to improve registration (line 303). What distinguishes this from Multi-resolution ICP as described by
Vlaminck, Michiel, Hiep Luong, and Wilfried Philips. "Multi-resolution ICP for the efficient registration of point clouds based on octrees." 2017 Fifteenth IAPR International Conference on Machine Vision Applications (MVA). IEEE, 2017.
Jost, Timothée, and Heinz Hugli. "A multi-resolution ICP with heuristic closest point search for fast and robust 3D registration of range images." Fourth International Conference on 3-D Digital Imaging and Modeling, 2003. 3DIM 2003. Proceedings. IEEE, 2003.
- To minimize noise and eliminate the background from the 3D point cloud, the authors used segmented NIR images. Did the author make an attempt to use the segmented RGB image?
- The title mentions the three-dimensional reconstruction of Rapeseed of the entire growing period. How did the reconstruction performance differ between stages? What are the difficulties associated with each stage?
Specific line comments are below
412 line It is unclear which algorithms are considered classic methods.
453 line It is unclear what constitutes a successful registration.
481 line (Section 4.4), It is unclear how the laser scans were denoised and recorded.
Lines 492-496, the text doesn’t reflect in Figure 9. It indicates laser data (9a) performed worse than the Kinect (9b)
Line 520, laser scanners are... challenging to fully automate. How a laser scanner is more difficult to automate than a Kinect.
Author Response
1: The multi-resolution registration algorithm in this document is similar in some ideas to this article. At first, it uses low-resolution point clouds to reduce the calculation time and increase the registration speed through downsampling, and then gradually increase it. Point cloud resolution. However, two parameters of the iterative registration in this article are changed at the same time, one is the size of the point cloud, that is, the point cloud resolution, which will affect the speed of point cloud registration; the other is the distance threshold of the matching point during registration, the distance threshold It is a very critical parameter, because the distance threshold is both the initial condition and the final result of registration, and its proper setting has a serious impact on the matching success rate and speed.
This document has been cited in the article and elaborated.
Point 2: To minimize noise and eliminate the background from the 3D point cloud, the authors used segmented NIR images. Did the author make an attempt to use the segmented RGB image?
Response 2: Color image segmentation is an important technique in phenotype detection. But in this article, the main goal is to obtain a complete 3D point cloud with accurate color information restoration. RGB-D alignment is to give color information to the depth map. If the task requires, color image segmentation can be performed after the alignment method in this article.
Point 3: The title mentions the three-dimensional reconstruction of Rapeseed of the entire growing period. How did the reconstruction performance differ between stages? What are the difficulties associated with each stage?
Response 3: The 3D reconstruction experiments of four growing season rapeseeds are supplemented and analyzed and explained in detail.
Point 4: 412 line It is unclear which algorithms are considered classic methods.
Response 4: The classic method is the traditional ICP registration algorithm, which only registers once, and does not change any parameters during the registration process. It has been supplemented in the text.
Point 5: 453 line It is unclear what constitutes a successful registration.
Response 5: Whether the registration is successful or not depends mainly on human eyes. The point cloud in this article is colored information, which makes observation easier. When the point cloud registration of the two perspectives is unsuccessful, it is easy to observe that the two point clouds are separated, rather than integrated. If the registration is successful, you need to calculate the quantitative data of the registration error in the program.
Point 6: 481 line (Section 4.4), It is unclear how the laser scans were denoised and recorded.
Response 6: We have added references in the text.
Point 7: Lines 492-496, the text doesn’t reflect in Figure 9. It indicates laser data (9a) performed worse than the Kinect (9b)
Response 7: The original text is incorrect and has been revised. In terms of spatial resolution and color reduction, the laser scanner surpasses the method in this paper, and the point cloud of the stem part exceeds the method in this paper, but the point cloud of the leaf and flower part where the curvature changes greatly. The algorithm in this paper surpasses the laser scanner.
Point 8: Line 520, laser scanners are... challenging to fully automate. How a laser scanner is more difficult to automate than a Kinect.

Round 2
Reviewer 1 Report
accept